# Designing Highly Efficient Cu_2_O-CuO Heterojunction CO Oxidation Catalysts: The Roles of the Support Type and Cu_2_O-CuO Interface Effect

**DOI:** 10.3390/nano12173020

**Published:** 2022-08-31

**Authors:** Fen Zhao, Yiyu Shi, Leilei Xu, Mindong Chen, Yingying Xue, Cai-E Wu, Jian Qiu, Ge Cheng, Jingxin Xu, Xun Hu

**Affiliations:** 1Collaborative Innovation Centre of the Atmospheric Environment and Equipment Technology, Jiangsu Key Laboratory of Atmospheric Environment Monitoring and Pollution Control, School of Environmental Science and Engineering, Nanjing University of Information Science & Technology, Nanjing 210044, China; 2College of Light Industry and Food Engineering, Nanjing Forestry University, Nanjing 210037, China; 3Jiangsu Shuangliang Environmental Technology Co., Ltd., Jiangyin 214400, China; 4State Environmental Protection Key Laboratory of Atmospheric Physical Modeling and Pollution Control, China Energy Science and Technology Research Institute Co., Ltd., Nanjing 210023, China; 5School of Material Science and Engineering, University of Jinan, Jinan 250022, China

**Keywords:** Cu_2_O-CuO heterojunction, support type, interface effect, CO oxidation

## Abstract

In this work, a series of Cu_2_O/S (S = α-MnO_2_, CeO_2_, ZSM-5, and Fe_2_O_3_) supported catalysts with a Cu_2_O loading amount of 15% were prepared by the facile liquid-phase reduction deposition–precipitation strategy and investigated as CO oxidation catalysts. It was found that the Cu_2_O/α-MnO_2_ catalyst exhibits the best catalytic activity for CO oxidation. Additionally, a series of Cu_2_O-CuO/α-MnO_2_ heterojunctions with varied proportion of Cu^+^/Cu^2+^ were synthesized by further calcining the pristine Cu_2_O/α-MnO_2_ catalyst. The ratio of the Cu^+^/Cu^2+^ could be facilely regulated by controlling the calcination temperature. It is worth noting that the Cu_2_O-CuO/α-MnO_2_-260 catalyst displays the best catalytic performance. Moreover, the kinetic studies manifest that the apparent activation energy could be greatly reduced owing to the excellent redox property and the Cu_2_O-CuO interface effect. Therefore, the Cu_2_O-CuO heterojunction catalysts supported on α-MnO_2_ nanotubes are believed to be the potential catalyst candidates for CO oxidation with advanced performance.

## 1. Introduction

With the acceleration of economic globalization, power plants, cement plants, automobile exhaust emissions [1,2], biomass combustion [3], and other sources of fuel produce large quantities of CO due to the incomplete combustion [4,5,6,7,8]. It is reported that when the CO content in the air is larger than 0.1%, it will cause poisoning in humans [9], which further results in nausea, dizziness, loss of consciousness, headache, and even fatal accidents [10]. As well known, CO is a colorless, odorless, and asphyxiating toxic gas, and a flammable and explosive air pollutant, which greatly threatens the health of humans and the safety of the living environment. There are various methods of CO removal reported in the literature, such as the physisorption, CO methanation [11], and catalytic oxidation [12]. Among these strategies, catalytic oxidation is regarded as one of the most efficient techniques for the elimination of CO [13,14]. Additionally, CO oxidation is widely investigated as an interesting probe reaction for other oxidation processes.

Currently, CO oxidation catalysts based on precious metals and transition metal oxides have attracted extensive research interest. Although the noble metal-based catalysts exhibit excellent low-temperature catalytic activity, their high cost and rarity limit their industrial applications. In comparison, metal oxide-based catalysts are much lower cost than the noble metal-based catalysts, which is more favorable for industrial applications [15,16,17]. For the transition metal oxides, Cu_2_O-based materials have been widely used as CO oxidation catalysts because of their high activity at low temperature [18]. These not only have the advantages of low cost, low toxicity, and easy synthesis process, but also have excellent redox performance. Therefore, they are considered to be the potential functionalized catalysts for CO oxidation. In recent years, the heterojunction of Cu_2_O with semiconductors, such as TiO_2_ [19], CeO_2_ [20], WO_3_ [21], BiVO_4_ [22], among others, has become a research focus [23]. Shi et al. found that CuO can provide lattice oxygen for the CO oxidation reaction, and then regenerate the oxygen atoms on the Cu_2_O surface through decomposition of O_2_ [7]. Both Cu_2_O and CuO are stable, abundant, low-cost, and environmentally friendly *p*-type semiconductors with direct band gaps of 2.2 and 1.2 eV, respectively. Copper oxides (Cu_2_O and CuO) have become important catalysts for photocatalytic degradation of organic pollutants due to their high light absorption coefficients [24,25]. The Cu_2_O and CuO composite structure has a synergetic effect on the low-temperature oxidation of CO. To be specific, Cu_2_O and CuO can provide active sites for the oxygen dissociation and CO oxidation, respectively, and the atomic-scale distance between Cu_2_O and CuO would be conducive to rapid migration of oxygen adsorption atoms on Cu_2_O-CuO [26]. Therefore, CuOx-based catalysts are potential candidates for the replacement of noble metal CO oxidation.

The CuOx-based catalysts supported on different supports have significant effects on CO oxidation. They can be divided into two categories depending on the reducibility of the support. As for the inert supports, generally a non-reducible oxide (Al_2_O_3_, SiO_2_, MgO, etc.), they mainly play the roles of dispersing and stabilizing active species. As for the reducible active supports, such as transition metal oxides (CeO_2_, CoO_x_, TiO_2_, etc.), they interact with metals and promote the reaction. Additionally, they are also partially involved in the reaction and contribute to the reaction activity. Specifically, CeO_2_ can form structurally stable and chemically active interfacial interactions with active components due to its unique oxygen storage and release capacity, which significantly improves the performance of the catalyst. Kong et al. [27] reported that the CeO_2_ catalyst could realize the efficient and stable removal of VOCs due to its high carbon deposition resistance. The ZSM-5 molecular sieve is one of the most important catalytic materials, which has been widely investigated as a support, catalyst, etc., in various fields [28,29,30]. Pang et al. [31] reported that the Ce-doped Cu/ZSM-5 catalyst greatly improved the catalytic performance, hydrothermal stability, and SO_2_ toxicity resistance in NH_3_-SCR selective catalytic reduction of NO. As for the Fe_2_O_3_, it has various advantages of good stability, low-cost, good oxygen carrier, and environmental friendliness. Zhao et al. [32] developed a Au/γ-Fe_2_O_3_ catalyst with commercial γ-Fe_2_O_3_ as the support, which exhibited 20 times higher activity for CO oxidation than the Au/α-Fe_2_O_3_ catalyst due to the higher redox property of Au/γ-Fe_2_O_3_. Additionally, the manganese oxide (MnOx), as an excellent catalyst support, has been attracting increasingly more attention due to its low price and environmentally harmless features. Mo et al. [33] prepared the CeO_2_/MnO_2_ catalyst with high efficiency for toluene degradation. Therefore, in order to further enhance the catalytic activity of CO oxidation at low temperature, it is of great significance to search for an optimal catalytic support for the CO oxidation catalyst.

In this work, a series of oxides, such as α-MnO_2_ nanotube, CeO_2_ nanosphere, ZSM-5, and commercial Fe_2_O_3_, were investigated as the supports for Cu_2_O-based CO oxidation catalysts. The optimal support was screened among these oxides by comparing their CO oxidation performance. The supports and catalysts were systematically characterized by X-ray powder diffraction (XRD), N_2_ physisorption, transmission electron microscopy (TEM), X-ray photoelectron spectroscopy (XPS), H_2_ temperature-programmed reduction (H_2_-TPR), among others. Additionally, the effect of the Cu_2_O-CuO heterojunction on CO oxidation performance was also studied in depth.

## 2. Experimental

### 2.1. Preparation of the Supports

The detailed preparation process of the α-MnO_2_, CeO_2_, and ZSM-5 supports are given in Appendix A.

### 2.2. Preparation of the Catalysts

#### 2.2.1. Preparation of Cu_2_O/S Supported Catalysts

A series of the Cu_2_O/S (S = α-MnO_2_, CeO_2_, ZSM-5, and Fe_2_O_3_) supported catalyst was prepared by the liquid-phase-reduction deposition–precipitation synthesis strategy. The specific process for Cu_2_O/S supported catalysts preparation is described in detail in Appendix A.

#### 2.2.2. Preparation of Cu_2_O-CuO Heterojunction Catalysts

The Cu_2_O-CuO/α-MnO_2_ heterojunction catalysts with various Cu^+^/Cu^2+^ ratios were synthesized. The Cu_2_O-CuO/α-MnO_2_ heterojunction catalyst obtained was designated as Cu_2_O-CuO/α-MnO_2_-T, in which “T” denotes the targeted calcination temperature. Further details related to preparation are shown in Appendix A.

### 2.3. Catalyst Characterizations

A series of characterization analyses were carried out on the supports and corresponding catalysts. Further details related to the equipment information, the operational details, and the determination parameters are summarized in Appendix A.

### 2.4. Catalytic Activity Measurements

The catalytic performance of catalyst for CO oxidation was evaluated in a fixed-bed reactor. The productions were detected online by gas chromatography. Further information on the specific reactor and evaluation of catalyst is summarized Appendix A.

## 3. Results and Discussions

### 3.1. Catalytic Property toward CO Oxidation

#### 3.1.1. Effect of the Support Type on the Catalytic Activity of CO Oxidation

The catalytic activity of CO oxidation on Cu_2_O/S (S = α-MnO_2_, CeO_2_, ZSM-5, Fe_2_O_3_) catalysts with different supports was evaluated to investigate the influence of the support type on the catalytic activity of the CO oxidation. Figure 1 shows that the trend of the CO conversions was increasing with the increase in reaction temperature until 100% CO conversion was finally reached. Additionally, it was also interesting to observe that the catalytic activities of the Cu_2_O-based catalysts supported on ZSM-5 and Fe_2_O_3_ were much lower than those of Cu_2_O/α-MnO_2_ and Cu_2_O/CeO_2_ catalysts. The presence of the α-MnO_2_ and CeO_2_ supports could greatly improve the catalytic activities of the Cu_2_O-based catalyst, which might be due to the excellent oxygen storage and release capacities of the α-MnO_2_ and CeO_2_ supports. This indicates that the properties of the supports had great effect on the catalytic performance of the Cu_2_O-based catalysts. It is worth noting that the Cu_2_O/α-MnO_2_ catalyst exhibited the best CO oxidation activity among the Cu_2_O-based catalysts investigated. Therefore, α-MnO_2_ was considered as the promising candidate among the investigated supports.

#### 3.1.2. Effect of Cu_2_O-CuO Heterojunction on the Catalytic Activity of CO Oxidation

As shown in Figure 2, detailed evaluations of the catalytic performance for CO oxidation on the Cu_2_O/α-MnO_2_, Cu_2_O-CuO/α-MnO_2_-T, and CuO/α-MnO_2_-500 catalysts were also conducted. Figure 2 indicates that the CO conversion on these catalysts increased with the increase in the reaction temperature until reaching 100%. In addition, all the investigated catalysts performed CO oxidation well, and 100% CO conversion could be achieved below 120 °C. However, their reaction temperatures of 10% (T_10_), 50% (T_50_) and 100% (T_100_) CO conversion were quite different. This might be due to the difference in Cu_2_O-CuO heterojunction composition between the CuO and Cu_2_O. Thus, the presence of the Cu_2_O-CuO heterojunction in Cu_2_O-CuO/α-MnO_2_-T catalyst could greatly decrease the ignition temperature (T_10_) of the catalyst. Specifically, the Cu_2_O-CuO/α-MnO_2_-T catalysts performed at lower ignition temperatures (even to the room temperature) than the pure Cu_2_O/α-MnO_2_ (73 °C) and CuO/α-MnO_2_-500 (48 °C). Furthermore, the T_50_ and T_100_ of the Cu_2_O-CuO/α-MnO_2_-T heterojunction catalysts were lower than those of the pure Cu_2_O/α-MnO_2_ and CuO/α-MnO_2_-500 catalysts. The phenomenon indicated that the Cu_2_O-CuO heterojunction of Cu_2_O-CuO/α-MnO_2_-T catalysts greatly contributed to enhanced CO oxidation activity at low temperature, which was on account of the synergetic effect of the Cu_2_O-CuO heterojunction [7,34]. The synergistic effect of Cu^+^ and Cu^2+^ was mainly derived from the atomic scale distance between Cu_2_O and CuO, which was conducive to the rapid migration of adsorbed oxygen on the Cu_2_O-CuO surface. In addition, it is noteworthy that the Cu_2_O-CuO/α-MnO_2_-260 catalyst with the lowest T_10_, T_50_, and T_100_ showed the highest reactivity among the Cu_2_O-CuO/α-MnO_2_-T catalysts investigated. Therefore, the reaction temperature required for the CO catalytic oxidation was greatly reduced and the efficient removal of CO could be realized at low temperature when assisted with the combined effect of the Cu_2_O-CuO heterojunction.

#### 3.1.3. Kinetic Study

The kinetic study of CO oxidation was carried out over the Cu_2_O/α-MnO_2_, Cu_2_O-CuO/α-MnO_2_-T, and CuO/α-MnO_2_-500 catalysts to investigate the Cu_2_O-CuO heterojunction on the catalytic performance. The Arrhenius curves are shown in Figure 3, and the specific value of the apparent activation energies are listed in Table 1. It is noteworthy that the apparent activation energies of the Cu_2_O-CuO/α-MnO_2_-T heterojunction catalysts were in the range from 41.9 to 62.2 kJ·mol^−1^, which were greatly lower compared with the pure Cu_2_O/α-MnO_2_ (83.4 kJ·mol^−1^) and CuO/α-MnO_2_-500 (64.0 kJ·mol^−1^) catalysts. These results indicate that the synergy effect of the Cu_2_O-CuO heterojunction can greatly increase the speed of the activation process of O_2_. Specifically, the Cu_2_O-CuO/α-MnO_2_-T catalyst reduced the activation energy of the CO oxidation process. At the same time, the apparent activation energy of Cu_2_O-CuO/α-MnO_2_-260 was the lowest among the Cu_2_O-CuO/α-MnO_2_-T catalysts. These results suggest that the Cu_2_O-CuO-T catalyst with the appropriate Cu^+^/Cu^2+^ ratio dramatically reduced the activation energy owing to the improvement of the O_2_ activation ability. The two rate-determining steps of the Cu_2_O-CuO/α-MnO_2_-260 °C catalysts were close to the optimal dynamic equilibrium ratio. Therefore, the Cu_2_O-CuO heterojunction structure displays great advantages in improving the CO oxidation activity at low temperature by reducing the apparent activation energy. In addition, it is noteworthy that the apparent activation energy of the CuO/α-MnO_2_-500 (64.0 kJ·mol^−1^) catalyst was also much lower than that of the Cu_2_O/α-MnO_2_ (83.4 kJ·mol^−1^) catalyst. This proved that the CuO-MnO_2_ interface provided the new reactive site for the CO catalytic oxidation, which was consistent with the study reported in [34]. The Cu_2_O-CuO/α-MnO_2_-T heterojunction catalysts performed excellent CO oxidation activity at low temperature through the comprehensive effects of the Cu_2_O-CuO heterojunction.

#### 3.1.4. Long-Term Stability Test

The Cu_2_O/α-MnO_2_, Cu_2_O-CuO/α-MnO_2_-260 and CuO/α-MnO_2_-500 catalysts were selected as representative catalysts for the CO oxidation stability test at 90 °C for 12 h under certain conditions. Their stability test results are displayed in Figure 4. It can be observed that these three catalysts showed excellent stability throughout the whole 12 h, with no signs of deactivation. Additionally, the Cu_2_O-CuO/α-MnO_2_-260 catalyst showed a higher conversion rate than the Cu_2_O/α-MnO_2_ and CuO/α-MnO_2_-500 catalysts in the catalytic stability test due to the presence of the Cu_2_O-CuO heterojunction. Similarly, the CuO/α-MnO_2_-500 catalyst showed higher conversion than the Cu_2_O/α-MnO_2_ catalyst. These results indicate that the Cu_2_O-CuO/α-MnO_2_-T heterojunction catalyst exhibited not only excellent low-temperature activity, but good stability owing to the synergistic interaction of Cu_2_O-CuO heterojunction.

### 3.2. Characterization of the Catalysts

#### 3.2.1. XRD Analysis

Appendix A displays the XRD patterns of the as-prepared Cu_2_O/S (S = α-MnO_2_, CeO_2_, ZSM-5, Fe_2_O_3_) catalysts with 15% Cu_2_O loading amount. As shown in Appendix A, the diffraction peaks of Cu_2_O/S catalysts were mainly situated at 2θ = 36.6°, 42.5°, 61.7°, and 73.7°, which could be ascribed to the (111), (200), (220), and (311) of Cu_2_O phase (PDF-#-05-0667). However, the intensities of the Cu_2_O peaks on these catalysts were quite different. Specifically, the intensities of the Cu_2_O peaks on the Cu_2_O/α-MnO_2_ and Cu_2_O/CeO_2_ catalysts were much weaker than that on the Cu_2_O/ZSM-5 and Cu_2_O/Fe_2_O_3_ catalysts, suggesting the high dispersion of the Cu_2_O species. This indicated that the properties of the support could greatly affect the dispersion of Cu_2_O, which was further conducive to the improvement of the catalytic activity of CO oxidation. In addition, the diffraction peaks of the supports could be observed over these Cu_2_O-based catalysts, which were attributed to the crystalline ZSM-5 (PDF-#-47-0715), Fe_2_O_3_ (PDF-#-99-0060), CeO_2_ (PDF-#-34-0394), and α-MnO_2_ (PDF-#-78-0428) phases.

Figure 5a shows the XRD patterns of the Cu_2_O/α-MnO_2_, Cu_2_O-CuO/α-MnO_2_-T, and CuO/α-MnO_2_-500 catalysts. The diffraction peaks of the Cu_2_O/α-MnO_2_ catalyst were mainly situated at 2θ = 36.6°, 42.5°, 61.7°, and 73.7°, which could be due to the (111), (200), (220), and (311) of the Cu_2_O phase (PDF-#-90-0041), respectively. The diffraction peaks of the CuO/α-MnO_2_-500 catalyst were located at 2θ = 35.7° and 38.8°, which might be due to the (−111) and (111) of the CuO phase (PDF-#-01-1117), respectively. However, as for the Cu_2_O-CuO/α-MnO_2_-T catalyst, the characteristic diffraction peaks of Cu_2_O, CuO, and α-MnO_2_ could also be found at the same time in their XRD patterns. The results indicate that the supported Cu_2_O had been transformed into the Cu_2_O-CuO heterojunction with a different proportion of Cu^+^/Cu^2+^. Specifically, the intensity of the CuO diffraction peaks increased with the increase in the calcination temperature. This suggests that the ratio of CuO in the Cu_2_O-CuO/α-MnO_2_-T catalyst increased with the increase in the calcination temperature from 240 to 280 °C. The brown Cu_2_O/α-MnO_2_ catalyst, abundant with the Cu_2_O phase, was eventually oxidized into the CuO/α-MnO_2_-500 black powder when the calcination temperature further increased to 500 °C. As a result, the XRD patterns of the CuO/α-MnO_2_-500 catalyst only displayed the characteristic diffraction peaks of CuO and α-MnO_2_ phases. This indicates that Cu_2_O was completely transformed into CuO after the calcination at 500 °C for 1 h. Based on this analysis, it can be concluded that the Cu_2_O-CuO/α-MnO_2_ heterojunction catalysts with different Cu^+^/Cu^2+^ ratios can be obtained by adjusting the calcination temperature. The appropriate Cu^+^/Cu^2+^ ratio heterojunction could greatly improve the catalytic activity of catalysts based on the catalytic results given in Figure 2.

Figure 5b shows the XRD patterns of the spent (SP-) Cu_2_O/α-MnO_2_, Cu_2_O-CuO/α-MnO_2_-260, and CuO/α-MnO_2_-500 catalysts after 12 h long-term stability tests. Meanwhile, the XRD patterns of their corresponding fresh catalysts are also presented for comparison. As shown in Figure 5b, this pattern represents the fresh catalyst, and the pattern below represents the spent catalyst (SP-) after the stability test. The characteristic diffraction peaks of Cu_2_O (PDF#-90-0041), CuO (PDF#-72-1982), and α-MnO_2_ (PDF#-78-0428) can still be obviously observed on the spent catalysts, especially on the SP-Cu_2_O-CuO/α-MnO_2_-260 heterojunction catalyst. Therefore, the XRD results confirmed that the Cu_2_O-CuO heterojunction existed in the Cu_2_O-CuO/α-MnO_2_-260 catalyst after the 12 h long-term stability test of CO oxidation. This implies that the oxygen activation cycle during the CO oxidation on the Cu_2_O-CuO/α-MnO_2_-260 heterojunction catalyst was sustainable owing to the thermal stability of the Cu_2_O-CuO heterojunction. At the same time, the characteristic peaks of α-MnO_2_ were also perfectly retained. These results illustrate the important roles of the Cu_2_O-CuO heterojunction interface when constructing stable and efficient CO oxidation catalysts.

#### 3.2.2. TG Analysis

The thermal stability and the phase transformation process of the Cu_2_O/α-MnO_2_ catalyst were studied by thermogravimetric analysis (TG) in air atmosphere. The weight decreased slowly in the range 30–200 °C, as shown in Figure 6. This might be caused by the removal of the water of physisorption and water of crystallization, together with the removal of trace organic reagent. However, when the temperature further rose to 511 °C, the weight began to drop sharply. The weight loss was equivalent to the loss of oxygen in the lattice of MnO_2_, leading to the formation of Mn_2_O_3_. Furthermore, when the temperature further increased to 769 °C, the weight again dropped sharply. This indicates that the Mn_2_O_3_ once again lost part of the lattice oxygen, leading to the formation of Mn_3_O_4_. These results are consistent with precursory literature reports [35,36,37]. It is worth noting that the mass loss of the catalyst (10.61%) was less than the theoretical value (12.26%) during the conversion of α-MnO_2_ to Mn_3_O_4_. The reason for the actual mass reduction being less than the theoretical mass reduction was that the transformation of the loaded Cu_2_O to CuO increased the weight.

#### 3.2.3. N_2_ Physisorption Analysis

The structural characteristics of the Cu_2_O/α-MnO_2_, Cu_2_O-CuO/α-MnO_2_-T, and CuO/α-MnO_2_-500 catalysts were investigated by N_2_ physisorption characterization. Their N_2_ adsorption–desorption isotherms and pore size distributions of catalysts are shown in Figure 7. It can be observed in Figure 7a that all the catalysts were characterized by the type IV isotherm and H_4_ hysteresis loops. This also indicates that the Cu_2_O-CuO/α-MnO_2_-T and CuO/α-MnO_2_-500 catalysts still had mesoporous structures similar to Cu_2_O/α-MnO_2_ after calcination at different temperatures, thus exhibiting the excellent thermal stability. Typically, the H_4_-shaped hysteresis loops indicate the existence of narrow wedge-shaped mesopores. Additionally, the mesopores might originate from the hollow α-MnO_2_ nanotube in the catalyst support, the sintering of catalysts in calcination process, and the rupture of internal mesopores. As shown in Figure 7b, the average pore size of the Cu_2_O-CuO/α-MnO_2_-T and CuO/α-MnO_2_-500 catalysts was similar or larger than the original Cu_2_O/α-MnO_2_ catalyst. This suggested that their mesoporous structures were not severely damaged by the thermal aggregation and phase transformation during the calcination process at different temperature. The nanotube hollow microsphere catalyst exhibited excellent thermal stability. In addition, the specific data of the structural properties of these catalysts are listed in Table 2. The results show that the specific surface areas, average pore diameters, and pore volumes of the Cu_2_O-CuO/α-MnO_2_-T catalysts were very similar to those of the pristine Cu_2_O/α-MnO_2_ catalyst. This again confirms the excellent thermal performance stability of these catalysts. The slight reduction in the specific surface area might be due to the sintering of Cu_2_O in the process of the calcination. It would further affect the surface morphology of the catalyst-supported nanotube hollow spheres by creating internal pores and surface defects. In contrast, the specific surface area of the CuO/α-MnO_2_-500 catalyst was relatively smaller, which might be caused by the complete oxidation of the surface Cu_2_O specifies and the closure and/or blockage of the hollow pores due to the long-term high temperature calcination.

#### 3.2.4. SEM and TEM Analyses

The TEM and SEM photos of the support α-MnO_2_ nanotube are shown in Appendix A. Figure 8 shows the SEM images of the as-prepared Cu_2_O/S (S = α-MnO_2_, CeO_2_, ZSM-5, Fe_2_O_3_) catalysts with 15% Cu_2_O loading amount. As shown in Figure 8a,b, the as-synthesized Cu_2_O/ZSM-5 catalyst exhibits regular cuboid shape with the a-axis (205 nm), b-axis (100 nm), and c-axis (1054 nm). In contrast, Cu_2_O/Fe_2_O_3_ catalyst supported on the commercial Fe_2_O_3_, shown in Figure 8c,d, resulted in an irregular morphology in the particle state, which might not be conducive to the dispersion of the Cu_2_O active sites. In addition, it is interesting to find in Figure 8e,f that the as-prepared Cu_2_O/CeO_2_ catalyst exhibited spherical nanoparticles with uniform size distribution about 130 nm. At the same time, it can be observed in Figure 8g,h that the Cu_2_O/α-MnO_2_ catalyst presented the morphology of hollow spheres, which were assembled by hollow nanotubes.

Figure 9 depicts the SEM images of the hollow Cu_2_O/α-MnO_2_ catalyst calcined at different temperatures. Further, the spatial distribution of the Cu and Mn elements in the catalysts were investigated by the EDS-mapping technique. The Cu_2_O/α-MnO_2_, Cu_2_O-CuO/α-MnO_2_-260, and CuO/α-MnO_2_-500 catalysts are chosen as the representative samples. As can be observed in Figure 9a,b, the Cu_2_O/α-MnO_2_ hollow microspheres catalyst with the average size of 5.01 μm exhibits the Cu_2_O nanoparticles supported on the surface of hollow nanotube. With the increase in calcination temperature, the surface of the Cu_2_O-CuO/α-MnO_2_-260 (Figure 9c,d) and CuO/α-MnO_2_-500 (Figure 9e,f) catalysts became coarse, and the morphology of hollow microsphere was less pronounced compared with the Cu_2_O/α-MnO_2_. This was because the Cu_2_O was gradually oxidized into CuO in the process of the calcination at different temperatures. In addition, the average particle size of the Cu_2_O-CuO/α-MnO_2_-260 (4.71 μm) and CuO/α-MnO_2_-500 (5.09 μm) hollow microspheres did not show significant thermal shrinkage and aggregation in the process of the calcination, and the morphology of hollow microspheres had been successfully maintained.

#### 3.2.5. FTIR Analysis

To investigate the phase transition process from Cu_2_O to CuO in the calcination process, FTIR analysis of Cu_2_O/α-MnO_2_, Cu_2_O-CuO/α-MnO_2_-T, and CuO/α-MnO_2_-500 catalysts in the range 400–4000 cm^−1^ was performed. As shown in Figure 10, it is noteworthy that the samples show infrared transmittance peaks at 598 and 468 cm^−1^, which are attributed to the stretching vibration of the Cu(+1)-O and Cu(+2)-O bond, respectively [38]. In addition, the coexistence of stretching vibration peaks at 598 and 468 cm^−1^ indicate the existence of the Cu_2_O-CuO heterojunction. When the calcination temperature increased to 240 °C or other higher temperature, the characteristic stretching vibration peak of Cu(+2)-O bond gradually appeared at 468 cm^−1^, while the Cu(+1)-O stretching vibration at 598 cm^−1^ progressively weakened due to the complete oxidation of Cu_2_O to CuO until it finally disappeared at 500 °C. This further verifies the change in crystal structure and valence state of Cu species in the calcination process in air atmosphere. Meanwhile, it can be found that the catalyst also provided strong infrared transmission peaks at 720 and 523 cm^−1^, which are attributed to the stretching vibration of O-Mn-O and layered manganese oxide, respectively [39,40,41,42]. As for the transmittance peaks around 3423 and 1637 cm^−1^, they are attributed to the stretching and flexural oscillations of the O-H groups caused by the physiosorbed water [43,44]. Therefore, the results of FTIR characterization further confirm the oxidation process of the Cu_2_O supported on the surface of α-MnO_2_ nanotube and the formation of Cu_2_O-CuO heterojunction in the process of the calcination.

#### 3.2.6. XPS Analysis

XPS analysis was performed on the Cu_2_O/S (S = ZSM-5, CeO_2_, α-MnO_2_, and Fe_2_O_3_) catalysts to further investigate the states of valence, surface chemical coordination, and composition. The results of Cu 2p and O 1s spectra are shown in Figure 11. Specifically, the pristine Cu_2_O/α-MnO_2_, Cu_2_O/Fe_2_O_3_, Cu_2_O/CeO_2,_ and Cu_2_O/ZSM-5 catalysts showed the 2p_3/2_ peak of Cu^+^ at 931.38 eV (Figure 11a). This indicates the presence of the Cu_2_O species on the support surface. As shown in Figure 11b, the O 1s main peak of the catalyst was around 529.36–529.46 eV, and the shoulder peak was around 530.76–530.96 eV. To be specific, the peaks of 529.36–529.46 eV and 530.76–530.96 eV could be attributed to surface adsorbed oxygen (O_ads_) and lattice oxygen (O_latt_), respectively [45]. However, the location of the main peak and shoulder peak of Cu_2_O/ZSM-5 catalyst differed greatly from the catalysts above. The shoulder peak at 530.58 eV should be attributed to the O species of Si-OH on the surface of SiO_2_ support, rather than the surface-adsorbed oxygen (O_latt_) [46,47]. In general, the number of the O_ads_ mainly depends on the number of oxygen vacancies at the surface. The reason is that the O_ads_ could only be absorbed on the oxygen vacancies. Thus, the O_ads_/(O_latt_ + O_ads_) ratio became the valid parameter to analyze the content of the surface oxygen vacancy. Table 3 shows that the O 1s shoulder peak area ratio (O_ads_/(O_latt_ + O_ads_)) of the Cu_2_O/CeO_2_, Cu_2_O/Fe_2_O_3_, Cu_2_O/ZSM-5, and Cu_2_O/α-MnO_2_ catalysts were 31.8%, 46.67%, 37.46%, and 21.26%, respectively. Although the (O_ads_/(O_latt_ + O_ads_)) ratio of Cu_2_O/α-MnO_2_ catalyst is somewhat lower compared with other supported Cu_2_O catalysts, the Cu_2_O/α-MnO_2_ catalyst exhibits excellent catalytic performance. This suggests that the ratio of O 1s shoulder peak area has a close relationship with the type of support.

The XPS results of Cu 2p, O 1s, and Mn 2p over Cu_2_O/α-MnO_2_, Cu_2_O-CuO/α-MnO_2_-T, and CuO/α-MnO_2_-500 catalysts are shown in Figure 12. Generally, there are two main peaks around 933.08–933.82 eV and 952.68–952.88 eV observed over these catalysts (Figure 12a), which can be attributed to the Cu 2p_3/2_ and Cu 2p_1/2_ peaks, respectively [48]. Interestingly, the Cu 2p_3/2_ peak was accompanied by an oscillating satellite peak from 944.78 to 945.28 eV. It was previously reported that the satellite peaks are generated by the transfer of electrons from the ligand orbital to the 3d orbital of Cu [49,50]. This indicates that the Cu^2+^ exists in the divalent form with the 3d^9^ structure, rather than the Cu^+^ or Cu^0^ species with the 3d^10^-filled energy level. It was reported that the CO oxidation activity of the Cu_2_O-CuO heterojunction catalyst is largely dependent on the surface of Cu^+^/Cu^2+^ ratio [26]. The Cu^2+^/Cu^1+^ relative percentages can be estimated by the peak-fitted areas of their corresponding XPS peaks. As shown in Table 4, the relative percentages of different oxidation states in the Cu_2_O/α-MnO_2_, Cu_2_O-CuO/α-MnO_2_-240, Cu_2_O-CuO/α-MnO_2_-260, Cu_2_O-CuO/α-MnO_2_-280, and CuO/α-MnO_2_-500 catalysts were 0, 6.30, 9.64, 9.99, and ∞, respectively. These results indicate that the calcination temperature greatly affects the relative percentages of different oxidation states over the catalyst surface. Meanwhile, the XPS results also reveal the formation of the Cu_2_O-CuO heterojunction structure on the surface of the Cu_2_O-CuO-T catalyst, which is consistent with the FTIR and XRD characterization results.

Figure 12a indicates that in the O 1s spectra of the catalysts studied, each catalyst showed a main peak centered at 529.40 eV and a shoulder peak around 531.12 eV. The results in 3 show that the O_ads_/(O_latt_ + O_ads_) ratio of the Cu_2_O-CuO/α-MnO_2_-T heterojunction catalyst was higher than that of Cu_2_O/α-MnO_2_. CuO/α-MnO_2_-500 was also found to have a relatively high acromion area ratio O_ads_/(O_latt_ + O_ads_), possibly due to the large amount of additional oxygen species provided by the CuO-MnO_2_ interface effect [34]. Among Cu_2_O-CuO-T heterojunction catalysts, Cu_2_O-CuO-260 catalyst possessed the highest proportion of shoulder peak area. According to previous report [51], the surface oxygen vacancies could enhance the redox properties of catalysts, which would be beneficial to the improvement of the performance of catalysts. The two binding energy peaks around 642.7 and 654.0 eV belong to Mn 2p_3/2_ and Mn 2p_1/2_ spin orbits, respectively, from the Mn 2p spectra shown in Figure 12c. It should be noted that these two peaks are the characteristic signals of Mn^4+^(IV). These phenomena indicate the occurrence of interfacial reactions and the existence of α-MnO_2_ [52,53].

#### 3.2.7. H_2_-TPR Analysis

To investigate the reduction, the Cu_2_O/S (S = ZSM-5, CeO_2_, α-MnO_2_, and Fe_2_O_3_), Cu_2_O/α-MnO_2_, Cu_2_O-CuO/α-MnO_2_-T, and CuO/α-MnO_2_-500 catalysts were performed H_2_-TPR analysis. The H_2_-TPR curves of the ZSM-5, CeO_2_, α-MnO_2_, and Fe_2_O_3_ supports are shown in Appendix A. As can be observed, ZSM-5 zeolite had no obvious reduction peak, which might be due to the inert support. Additionally, it could be interesting to find that the Fe_2_O_3_ support exhibited a small reduction peak at 532 °C and a wide non-termination peak at 745 °C. According to the literature [54], the Fe_2_O_3_ support experiences the reduction from Fe_2_O_3_ to Fe_3_O_4_ and from Fe_3_O_4_ to Fe. The reduction of CeO_2_ support might be due to the existence of Ce^4+^/Ce^3+^ redox pairs. In addition, α-MnO_2_ showed two reduction peaks at 432 and 579 °C. The reduction process of α-MnO_2_ samples could be speculated to be a two-step reduction process with MnO as the final state, namely MnO_2_→Mn_3_O_4_→MnO [34,44,55].

As for the H_2_-TPR curves of Cu_2_O/S catalysts supported on different oxides, shown in Figure 13a, they reveal two or three reduction peaks in the range 30–800 °C. This indicates that the reduction of Cu_2_O species at different temperatures was related to the different metal–support interaction. Specifically, the Cu_2_O/ZSM-5 catalyst showed a large reduction peak and a shoulder peak near 296 and 420 °C, respectively, which could correspond to the interaction strength of the Cu_2_O and ZSM-5 with different intensities. The Cu_2_O/Fe_2_O_3_ catalyst showed three reduction peaks, which might correspond to the interaction strength of Cu_2_O and Fe_2_O_3_ with different intensities, and the reduction peak of Fe_2_O_3_ support. The Cu_2_O/CeO_2_ catalyst showed two reduction peaks, and the peak at 258 °C might be the common reduction peak of Cu_2_O and CeO_2_ species. It is worth noting that the reduction peak intensity of the Cu_2_O/α-MnO_2_ catalyst was the largest among these four catalysts. Therefore, the nature of the support had an important influence on the reduction property of the catalyst and the metal–support interaction.

The H_2_ reduction profiles of the Cu_2_O/α-MnO_2_, Cu_2_O-CuO/α-MnO_2_-T, and CuO/α-MnO_2_-500 catalysts are shown in Figure 13b. These catalysts had three reduction peaks: 156–290, 337–385, and 442–487 °C. Generally, their H_2_-TPR curves are analogous in the shape, with a large reduction peak at 442–487 °C (γ-type), a big shoulder peak at 337–385 °C (β-type), and a small shoulder peak at 156–290 °C (α-type). Additionally, it is noteworthy that the reduction peak area of the CuO/α-MnO_2_-500 catalyst was somewhat larger than that of the Cu_2_O/α-MnO_2_ catalyst. The main reason for this was that the reduction process of the CuO consisted of two reduction stages, which contained the reduction processes from Cu^2+^ to Cu^+^ and then from Cu^+^ to Cu^0^. Therefore, compared with the reduction of Cu_2_O/α-MnO_2_, the reduction of the Cu_2_O-CuO/α-MnO_2_ heterojunction and CuO/α-MnO_2_ catalysts required more H_2_ reductant, resulting in the larger H_2_ consumption. Furthermore, it is interesting to find that the initial reduction temperature of Cu_2_O-CuO/α-MnO_2_-T heterojunction catalysts gradually shifted to the lower temperature with the increase in the calcination temperature. This suggests that the presence of the Cu_2_O-CuO heterojunction had an important influence on the reduction property of the catalysts.

## 4. Conclusions

In summary, a series of Cu_2_O/S (S = α-MnO_2_, CeO_2_, ZSM-5, Fe_2_O_3_) catalysts were prepared by the liquid-phase reduction deposition–precipitation strategy and used for CO oxidation. It was found that the Cu_2_O/α-MnO_2_ catalyst performed the optimum low-temperature activity of CO oxidation. Furthermore, the influence of the Cu_2_O-CuO heterojunction structure on the catalytic activity of CO oxidation was also carefully investigated. It was found that Cu_2_O-CuO/α-MnO_2_-260 with the Cu_2_O/total Cu proportion of 9.4% exhibited the highest catalytic activity. The presence of the Cu_2_O-CuO heterojunction greatly increased the content of the surface oxygen vacancy. This further enhanced the activation ability of oxygen, and finally improved the low temperature CO oxidation property. Kinetic study showed that the Cu^+^/Cu^2+^ proportion of the Cu_2_O-CuO heterojunction and redox property of the Cu_2_O-CuO/α-MnO_2_-T catalyst significantly reduced the apparent activation energy of CO oxidation. As a result, the catalytic activity of CO oxidation at low temperature was greatly improved. In addition, there are few reports on Cu_2_O-CuO heterojunction catalysts supported on MnO_2_ for CO oxidation. Although researchers have prepared CuOx-type catalysts with MnO_2_ as the support by some methods, its catalytic CO oxidation performance is not ideal. In conclusion, the Cu_2_O-CuO/α-MnO_2_-T heterojunction catalysts with adjustable Cu^+^/Cu^2+^ ratios are expected to be promising catalyst candidates for CO oxidation in future applications.

## Figures and Tables

**Figure 1 nanomaterials-12-03020-f001:**
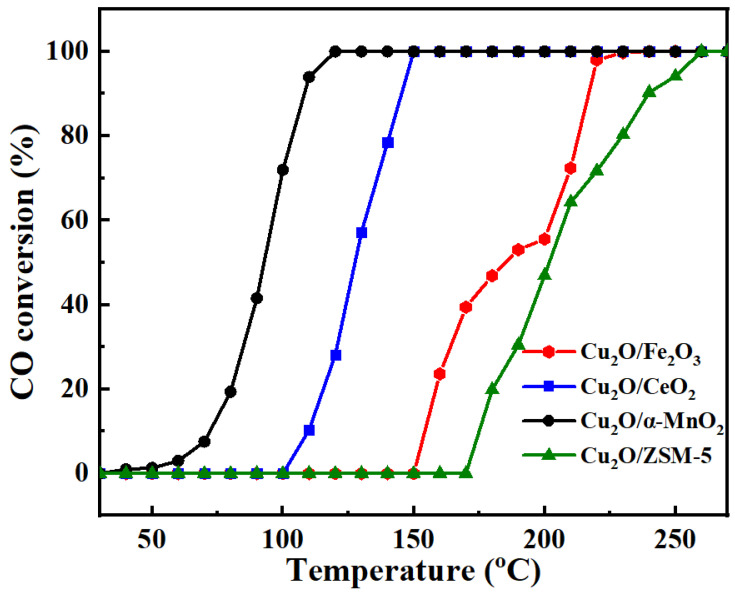
The curves of CO conversion versus reaction temperature over the as-prepared Cu_2_O/S (S = ZSM-5, CeO_2_, α-MnO_2_, and Fe_2_O_3_) catalysts; reaction conditions: CO/O_2_/N_2_ = 1/20/79, GHSV = 12,000 mL g^−1^ h^−1^, 1 atm.

**Figure 2 nanomaterials-12-03020-f002:**
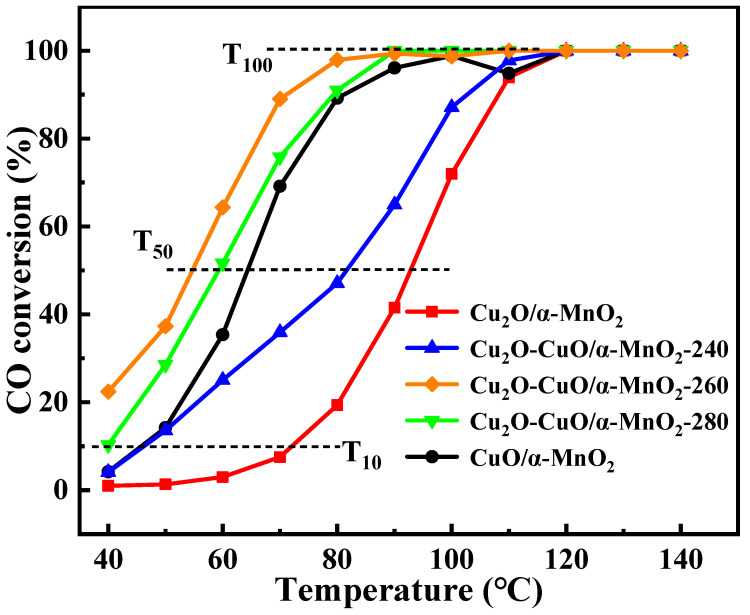
The curves of the CO conversion versus reaction temperature over the as-prepared Cu_2_O/α-MnO_2_, Cu_2_O-CuO/α-MnO_2_-T, and CuO/α-MnO_2_-500 catalysts; reaction conditions: CO/O_2_/N_2_ = 1/20/79, GHSV = 12,000 mL g^−1^ h^−1^, 1 atm.

**Figure 3 nanomaterials-12-03020-f003:**
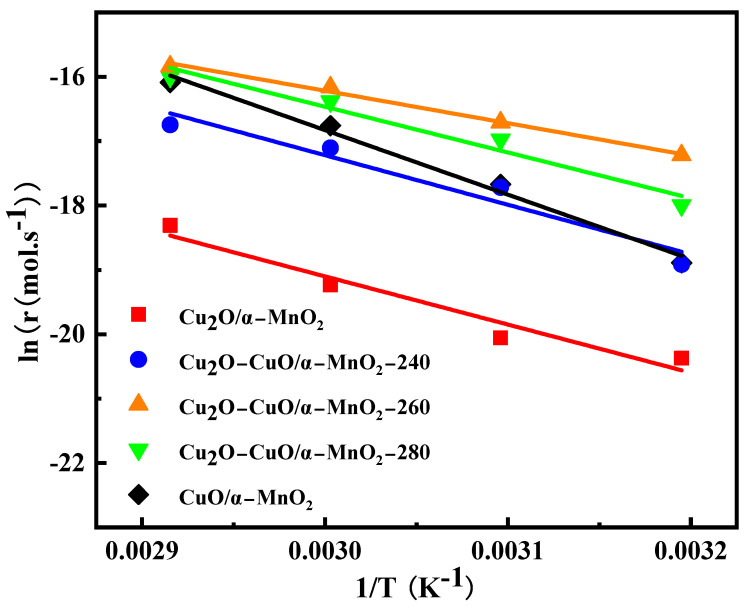
Arrhenius plots for the CO oxidation reaction rate over the as-prepared Cu_2_O/α-MnO_2_, Cu_2_O-CuO/α-MnO_2_-T, and CuO/α-MnO_2_-500 catalysts.

**Figure 4 nanomaterials-12-03020-f004:**
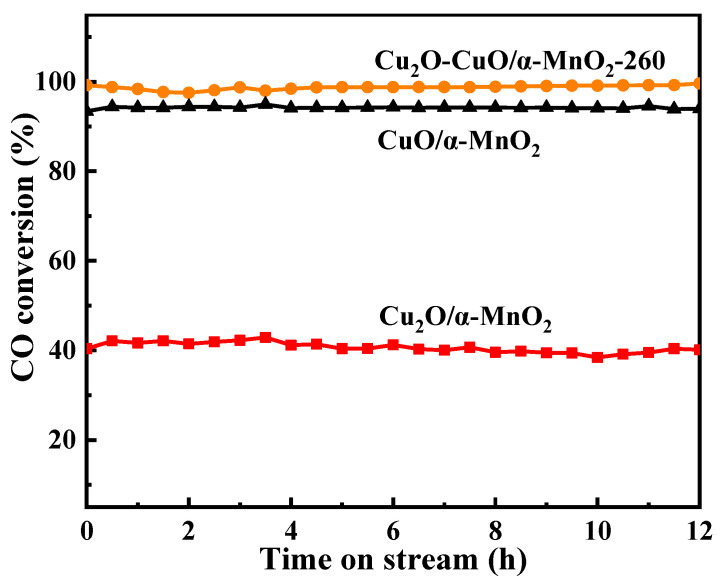
Results of 12 h long-term stability tests over the as-prepared Cu_2_O/α-MnO_2_, Cu_2_O-CuO/α-MnO_2_-T, and CuO/α-MnO_2_-500 catalysts; reaction conditions: CO/O_2_/N_2_ = 1/20/79, GHSV = 12,000 mL g^−1^ h^−1^, 1 atm.

**Figure 5 nanomaterials-12-03020-f005:**
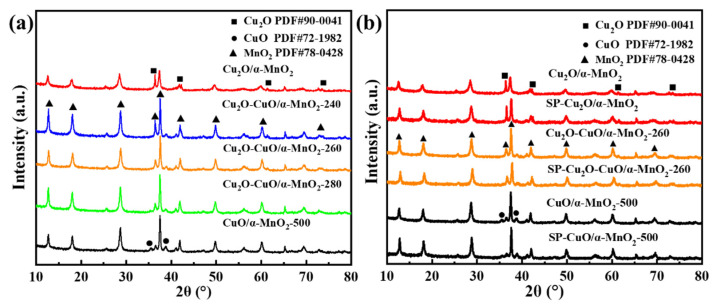
X-ray diffraction patterns of the as-prepared (**a**) Cu_2_O/α-MnO_2_, Cu_2_O-CuO/α-MnO_2_-T, and CuO/α-MnO_2_-500 catalysts and (**b**) the XRD patterns of each catalyst after long-term stability tests (SP-).

**Figure 6 nanomaterials-12-03020-f006:**
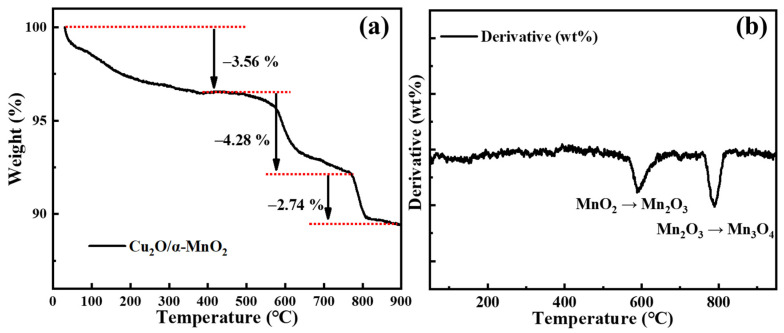
(**a**) TG result and (**b**) the DTG result for the Cu_2_O/α-MnO_2_ in air atmosphere.

**Figure 7 nanomaterials-12-03020-f007:**
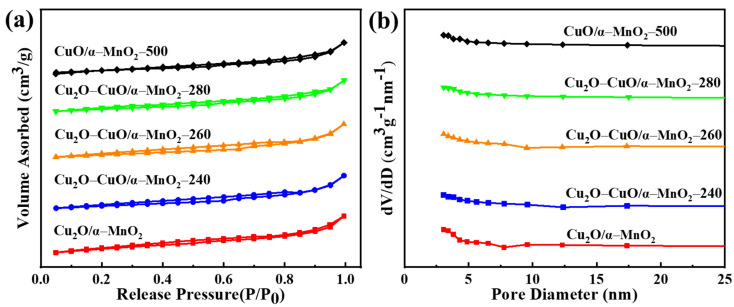
(**a**) N_2_ adsorption–desorption isotherms and (**b**) pore size distribution curves of the as-prepared Cu_2_O/α-MnO_2_, Cu_2_O-CuO/α-MnO_2_-T, and CuO/α-MnO_2_-500 catalysts.

**Figure 8 nanomaterials-12-03020-f008:**
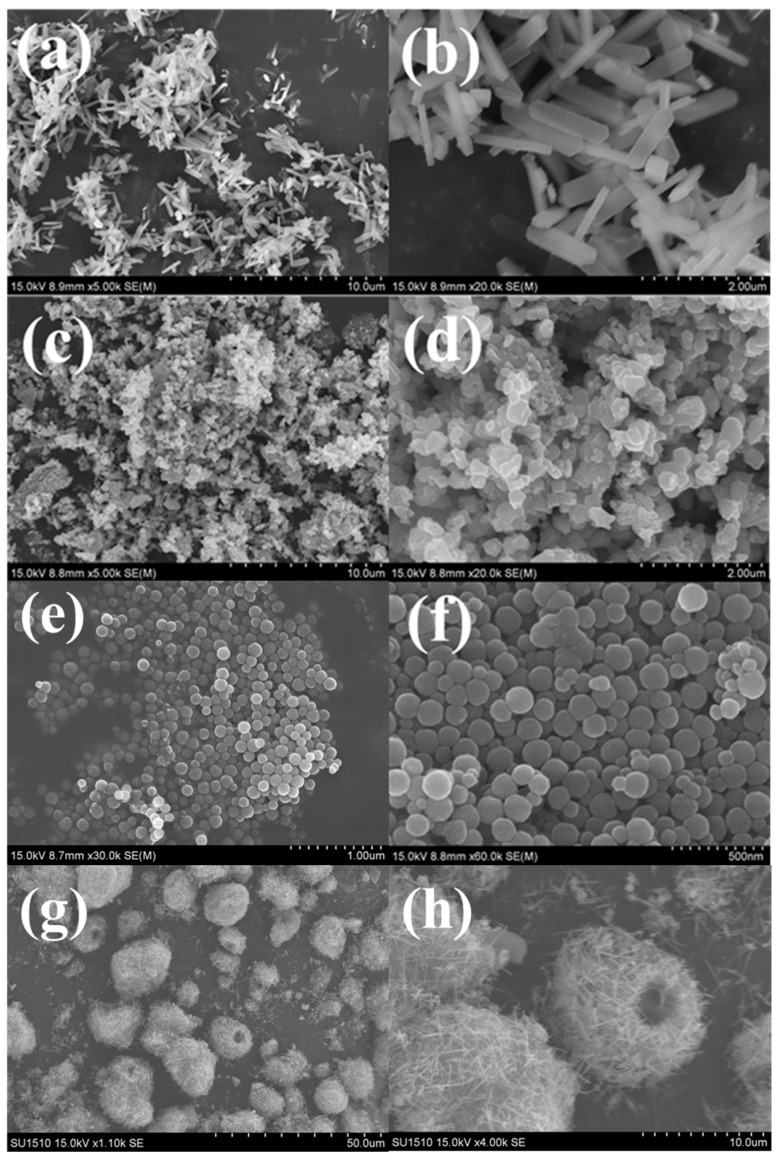
SEM images of the as-prepared (**a**,**b**) Cu_2_O/ZSM-5, (**c**,**d**) Cu_2_O/Fe_2_O_3,_ (**e**,**f**) Cu_2_O/CeO_2_, and (**g**,**h**) Cu_2_O/α-MnO_2_ catalysts.

**Figure 9 nanomaterials-12-03020-f009:**
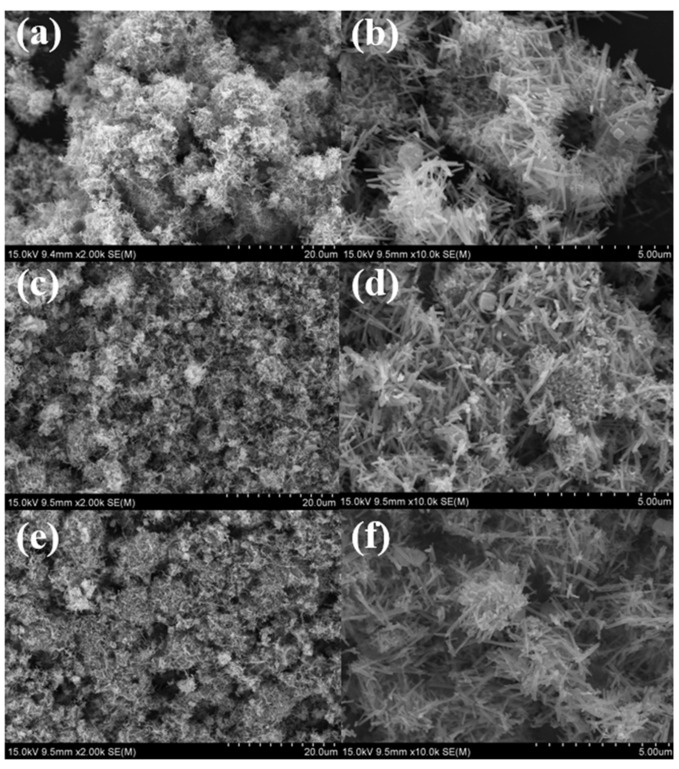
SEM images of the as-prepared catalysts: (**a**,**b**) Cu_2_O/α-MnO_2_, (**c**,**d**) Cu_2_O-CuO/α-MnO_2_-260, and (**e**,**f**) CuO/α-MnO_2_-500.

**Figure 10 nanomaterials-12-03020-f010:**
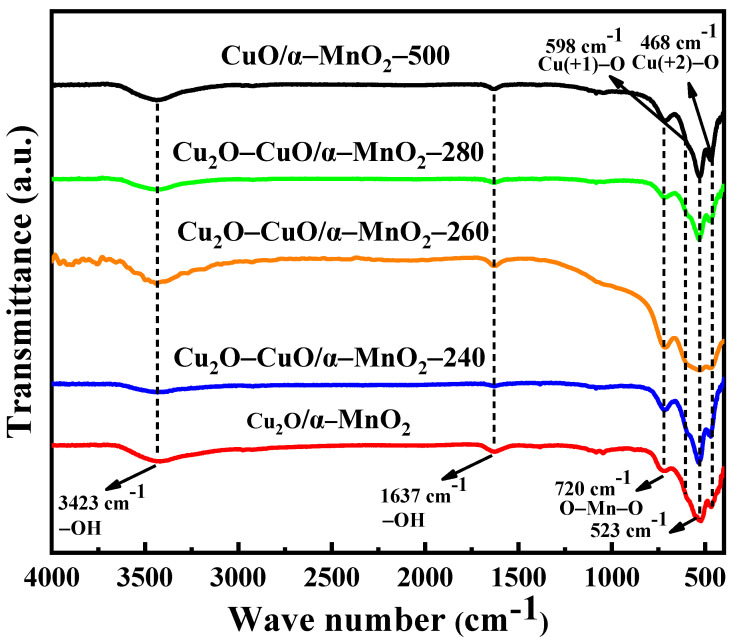
FTIR characterization for the as-prepared Cu_2_O/α-MnO_2_, Cu_2_O-CuO/α-MnO_2_-T, and CuO/α-MnO_2_-500 catalysts.

**Figure 11 nanomaterials-12-03020-f011:**
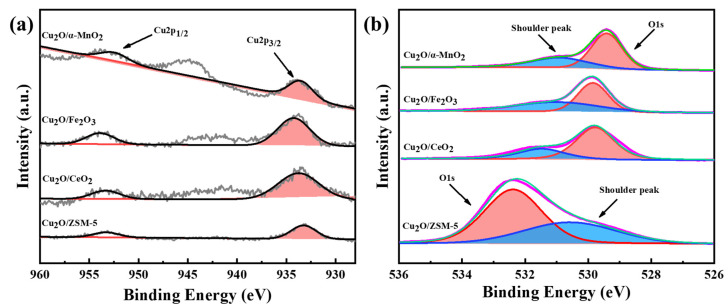
(**a**) Cu 2p and (**b**) O 1s XPS spectra of the as-prepared Cu_2_O/S (S = ZSM-5, CeO_2_, α-MnO_2_, and Fe_2_O_3_) catalysts.

**Figure 12 nanomaterials-12-03020-f012:**
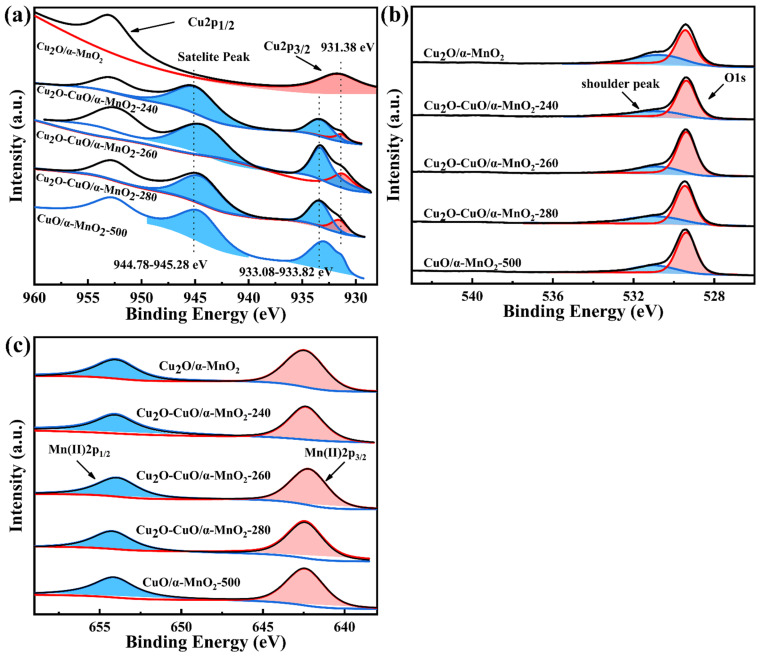
(**a**) Cu 2p, (**b**) O 1s, and (**c**) Mn 2p XPS spectra of the as-prepared Cu_2_O/α-MnO_2_, Cu_2_O-CuO/α-MnO_2_-T, and CuO/α-MnO_2_-500 catalysts.

**Figure 13 nanomaterials-12-03020-f013:**
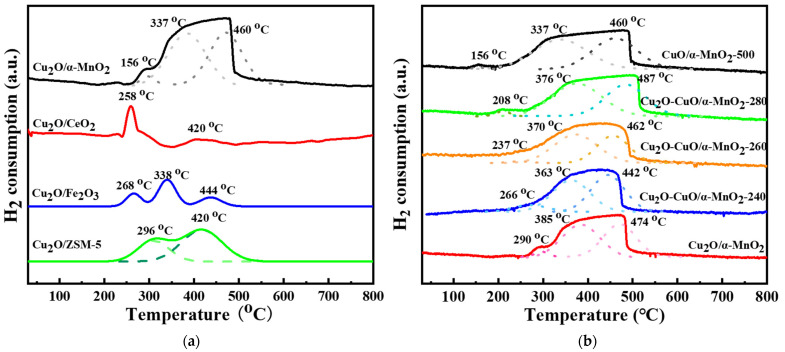
H_2_-TPR profiles of the as-prepared (**a**) Cu_2_O/S (S = ZSM-5, CeO_2_, α -MnO_2_, and Fe_2_O_3_) catalysts and (**b**) α-MnO_2_, Cu_2_O/α-MnO_2_, Cu_2_O-CuO/α-MnO_2_-T, and CuO/α-MnO_2_-500 catalysts.

**Table 1 nanomaterials-12-03020-t001:** Apparent activation energy data of the as-prepared Cu_2_O/α-MnO_2_, Cu_2_O-CuO/α-MnO_2_-T, and CuO/α-MnO_2_-500 catalysts.

Samples	Ea (KJ/mol)
**Cu_2_O/α-MnO_2_**	83.4
**Cu_2_O-CuO/α-MnO_2_-240**	62.2
**Cu_2_O-CuO/α-MnO_2_-260**	41.9
**Cu_2_O-CuO/α-MnO_2_-280**	59.2
**CuO/α-MnO_2_-500**	64.0

**Table 2 nanomaterials-12-03020-t002:** Structural properties of the as-prepared catalysts.

Samples	Specific Surface Area (m^2^/g)	Pore Volume (cm^3^/g)	Average Pore Diameter (nm)	Isotherm Type
**Cu_2_O/α-MnO_2_**	43.64	0.051	3.05	IV H4
**Cu_2_O-CuO/α-MnO_2_-240**	28.33	0.048	3.06	IV H4
**Cu_2_O-CuO/α-MnO_2_-260**	28.82	0.047	3.06	IV H4
**Cu_2_O-CuO/α-MnO_2_-280**	26.37	0.045	3.06	IV H4
**CuO/α-MnO_2_-500**	10.49	0.045	3.06	IV H4

**Table 3 nanomaterials-12-03020-t003:** O 1s peak area of the as-prepared Cu_2_O/S (S = ZSM-5, CeO_2_, α-MnO_2_, and Fe_2_O_3_) Cu_2_O/α-MnO_2_, Cu_2_O-CuO/α-MnO_2_-T and CuO/α-MnO_2_-500 catalysts.

Samples	O 1s Main Peak Area	O 1s Shoulder Peak Area	O 1s Shoulder Peak Area Ratio ^a^ (%)
**Cu_2_O/CeO_2_**	283,829.4	132,367.2	31.80
**Cu_2_O/Fe_2_O_3_**	173,370.0	151,701.6	46.67
**Cu_2_O/ZSM-5**	653,388.9	391,390.7	37.46
**Cu_2_O/α-MnO_2_**	223,908.2	60,458.9	21.26
**Cu_2_O-CuO/α-MnO_2_-240**	223,952	88,670.38	28.36
**Cu_2_O-CuO/α-MnO_2_-260**	222,799	123,070.98	35.58
**Cu_2_O-CuO/α-MnO_2_-280**	227,844.3	119,566.3	34.42
**CuO/α-MnO_2_-500**	248,099.8	98,654.84	28.45

^a^ The O 1s shoulder peak area ratio represents the ratio of O_ads_/(O_latt_ + O_ads_).

**Table 4 nanomaterials-12-03020-t004:** Cu 2p peak area of the as-prepared Cu_2_O/S (S = α-MnO_2_, CeO_2_, SBA-ZSM-5, Fe_2_O_3_), Cu_2_O/α-MnO_2_, Cu_2_O-CuO/α-MnO_2_-T, and CuO/α-MnO_2_-500 catalysts.

Samples	Cu_2_O Peak Area	CuO Peak Area	Satellite Peak Area	Cu_2_O Peak Area Percentage (%)	CuO Peak Area Percentage (%)	Cu^2+^/Cu^1+^ Relative Ratio
**Cu_2_O/CeO_2_**	58,985.93	0	0	100.0	0	0
**Cu_2_O/Fe_2_O_3_**	23,117.54	0	0	100.0	0	0
**Cu_2_O/ZSM-5**	15,803.46	0	0	100.0	0	0
**Cu_2_O/α-MnO_2_**	28,836.83	0	0	100.0	0	0
**Cu_2_O-CuO/α-MnO_2_-240**	8059.77	16,647.28	33,992.70	13.7	86.3	6.30
**Cu_2_O-CuO/α-MnO_2_-260**	8803.58	36,178.22	49,118.25	9.4	90.6	9.64
**Cu_2_O-CuO/α-MnO_2_-280**	7076.77	16,647.28	54,040.24	9.1	90.9	9.99
**CuO/α-MnO_2_-500**	0	55,346.21	96,344.52	0.0	100	∞

## Data Availability

The data supporting the findings of this study are available by reason-able request to leileixu88@gmail.com.

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
