# Peer review of "Designing Highly Efficient Cu2O-CuO Heterojunction CO Oxidation Catalysts: The Roles of the Support Type and Cu2O-CuO Interface Effect"

_nanomaterials, 2022, doi:10.3390/nano12173020_

Round 1
Reviewer 1 Report
Zhao et al have conducted a study on a series of Cu2O/S (S = α-MnO2, CeO2, ZSM-5, and Fe2O3) supported catalysts for CO where Cu2O loading amount are fixed at 15%. They have synthesized these materials via liquid phase reduction deposition precipitation strategy and examined its CO oxidation catalysts property. They carried out extensive study and found the best CO catalytic activity for Cu2O/α-MnO2 among all. Furthermore, they have also calcined the sample and claim the formation of Cu2O-CuO/α-MnO2 heterojunction with varied proportion of Cu+/Cu2+. They have also claimed via kinetic study that the activation energy could be greatly reduced owing to the excellent redox property and the Cu2O-CuO interface effect. In my opinion, this paper can be accepted after addressing following comments:
1. Authors claim the formation of interface between Cu2O-CuO/α-MnO2 heterojunction. Provide the proper energy band diagram for this.
2. Authors reported the maximum catalytic activity for Cu2O/α-MnO2 among all. Surface area also play important role in catalytic activity. How do authors normalize the surface area of different morphology?
3. In Fig. 8. Magnification of all SEM image are not same. Why?
4. In Fig 12 (a), assign Cu (1+) and Cu(2+) with binding energy.
5. In supporting information, Authors have presented only TEM analysis of MnO2, not others. Why?
Author Response
Zhao et al have conducted a study on a series of Cu2O/S (S = α-MnO2, CeO2, ZSM-5, and Fe2O3) supported catalysts for CO where Cu2O loading amount are fixed at 15%. They have synthesized these materials via liquid phase reduction deposition precipitation strategy and examined its CO oxidation catalysts property. They carried out extensive study and found the best CO catalytic activity for Cu2O/α-MnO2 among all. Furthermore, they have also calcined the sample and claim the formation of Cu2O-CuO/α-MnO2 heterojunction with varied proportion of Cu+/Cu2+. They have also claimed via kinetic study that the activation energy could be greatly reduced owing to the excellent redox property and the Cu2O-CuO interface effect. In my opinion, this paper can be accepted after addressing following comments:
Answer: Thank you very much for providing such constructive comments and valuable suggestions to us. Your valuable comments are greatly important to improve the quality of manuscript. The authors have made the best efforts to revise the whole manuscript according to your valuable and inspiring comments. Please kindly check the revised manuscript as the reference. The authors sincerely wish that the revised manuscript and the corresponding explanations could make you satisfied. Any further suggestions and comments are also greatly welcome and highly appreciated. Thank you.
- Authors claim the formation of interface between Cu2O-CuO/α-MnO2 heterojunction. Provide the proper energy band diagram for this.
Answer: Thank you very much for your inspiring and sensible suggestion. In fact, the authors also plan to add energy band diagram. However, the UV-VIS results cannot be available soon due to the close of the lab in summer holiday during the background of the pandemic COVID-19. The authors sincerely apologize for this. Thank you very much for your understanding. In addition, the Cu2O-CuO heterojunction catalyst was characterized by UV-VIS in our previous study (Ind. Eng. Chem. 2022; 105:324-336). It was found that the band gap of Cu2O-CuO heterojunction was significantly narrower than that of pure Cu2O and CuO counterpart. In addition, the effective band bending also confirmed the formation of Cu2O-CuO heterojunctions in these Cu2O-CuO heterojunction catalysts. The authors sincerely wish that the explanation could make you satisfied and clear. Thank you.
- Authors reported the maximum catalytic activity for Cu2O/α-MnO2 among all. Surface area also play important role in catalytic activity. How do authors normalize the surface area of different morphology?
Answer: Thank you very much for your constructive comment. Your concern is quite reasonable. The authors studied the influences of the support and Cu2O-CuO heterojunction on the catalytic activity of CO oxidation. Firstly, the optimal support was screened and selected from the four specified supports. Then, the effect of Cu2O-CuO heterojunction on the catalytic activity was investigated. Among them, the properties of the support were considered comprehensively, including its specific surface area and redox properties. The surface area was also brought into the discussion considering inherent nature as different supports. The authors sincerely wish that our explanation can make you satisfied and clear. Thank you.
- In Fig. 8. Magnification of all SEM image are not same. Why?
Answer: Thank you very much for your attention to our manuscript. This manuscript focused on the influence of two aspects on the catalytic activity of CO: one was the support, and the other was Cu2O-CuO heterojunction. Among them, the author selected the most representative four supports after careful consideration to explore their influence on the catalytic performance of Cu2O materials. All the characteristics of the supports were discussed as their inherent attributions. The morphology and size of the four designated supports were different, but they were of great research value. In order to better understand the morphology and size of the support, the SEM images were taken in different magnifications. The authors sincerely wish that our explanation can make you satisfied and clear. Thank you.
- In Fig 12 (a), assign Cu (1+) and Cu (2+) with binding energy.
Answer: Thank you very much for your constructive and sensible suggestion. Following your advice, the authors have assigned Cu (1+) and Cu (2+) with binding energy in Fig 12 (a). Please kindly find the revised manuscript as the reference. Thank you.
- In supporting information, Authors have presented only TEM analysis of MnO2, not others. Why?
Answer: Thank you very much for your attention to our manuscript. Your comments are greatly reasonable. In this work, a series of oxides, such as α-MnO2 nanotube, CeO2 nanosphere, ZSM-5, and commercial Fe2O3, were investigated as the supports for Cu2O-based CO oxidation catalysts. The optimal support would be screened among these oxides by comparing their CO oxidation performance. From the point of view of catalytic performance, the Cu2O/α-MnO2 catalyst showed the best catalytic performance of CO oxidation. Therefore, the authors believed that the Cu2O-CuO heterojunction could be better studied by selecting the most promising α-MnO2 support for further characterization. Therefore, the authors showed TEM images of α-MnO2 in the supporting information to better observe its nanotube structure. Actually, the morphologies of the other supports could be observed in Figure 8. Currently, the authors are sincerely sorry that the TEM images cannot be provided in the revised manuscript due to the close of the lab during the COVID-19 background and the summer vacation. Besides, the authors will further characterize other supports in the future work. Thank you so much for your kind understanding.

Reviewer 2 Report
This work examined the structures of a series of CuOx/Metal Oxide catalysts with different metal oxide supports (α-MnO2, CeO2, ZSM-5, and Fe2O3). Cu deposited on α-MnO2 was found to show higher activity. In this work, the authors proposed that the catalytic performance can be efficiently enhanced by the redox property and the Cu2O-CuO interface affected the catalytic performance for CO oxidation. The subject is definitely of particular importance both from the fundamental and practical points of view toward the rational design of catalysts. I consider, however, that various major issues listed below hinder the publication of the article in the present form.
- The catalytic performance of Cu deposited on α-MnO2, ie., the best catalyst in this manuscript, should be compared with the reported catalysts in the open literature, which will enhance the value of the whole work.
- On page 2, there is a different font style has been used. Authors should use the same font style throughout the manuscript.
- Authors show the XPS peak fitting areas in Table 3 and Table 4. However, it does not give any meaningful information. It’s better to calculate the relative percentages of different oxidation states.
- How were the dispersion patterns of CuOx on these supports? Atomically dispersed? Or in the form of clusters or particles? The dispersion of Cu should be measured in order to support the activity.
- It is very surprising to see that there is no XPS Cu 2p ½ peak in Cu2O/α-MnO2 catalysts (Fig. 11 and Fig. 12). This is completely wrong. Authors should repeat the XPS analysis of Cu 2p in all samples.
- Authors should record the XPS Mn 3p to differentiate the Mn oxidation states. The current Mn 2p spectra do not give any meaningful information.
- It is well known that the discrimination of different Cu oxidation states by XPS cannot be easily performed; a comparison with Auger parameters should be performed (Catal. Sci. Technol., 2021, 11, 6134-6142).
- On page 14, very small labels in Figure 12 are difficult to read, and the figure quality needs to be improved.
- The title speaks about “The Roles of the Support Type and Cu2O-CuO Interface Effect”. There is no evidence of the involvement of the Cu2O-CuO interface effect.
Author Response
This work examined the structures of a series of CuOx/Metal Oxide catalysts with different metal oxide supports (α-MnO2, CeO2, ZSM-5, and Fe2O3). Cu deposited on α-MnO2 was found to show higher activity. In this work, the authors proposed that the catalytic performance can be efficiently enhanced by the redox property and the Cu2O-CuO interface affected the catalytic performance for CO oxidation. The subject is definitely of particular importance both from the fundamental and practical points of view toward the rational design of catalysts. I consider, however, that various major issues listed below hinder the publication of the article in the present form.
Answer: Thank you very much for your selfless dedication and providing such inspiring, constructive, and sensible comments for the authors. Your valuable comments are of great significance to improve the quality of the manuscript. Following your valuable suggestions, the authors have devoted our best efforts to revising the whole manuscript. The authors sincerely wish that the revised manuscript can make you satisfied. Please kindly check the revised manuscript as the reference. Any further suggestions as well as comments are also greatly welcome and appreciated.
- The catalytic performance of Cu deposited on α-MnO2, ie., the best catalyst in this manuscript, should be compared with the reported catalysts in the open literature, which will enhance the value of the whole work.
Answer: Thank you very much for your constructive comments. According to your comments, the authors reviewed relevant literatures and compared the Cu2O-CuO/α-MnO2-260 catalysts with those reported in the same field. The authors found that there are few reports on Cu2O-CuO heterojunctions catalysts supported on MnO2 for CO oxidation. Although researchers have prepared CuOx-type catalysts with MnO2 as the support by some methods, its catalytic oxidation performance of CO is not ideal (Reac Kinet Mech Cat. 2013; 108: 173–182.) (J. Environ. Chem. Eng. 2017; 5: 5179-5186.) (Appl. Surf. Sci. 2013; 273: 357-363). As a result,the Cu2O-CuO/α-MnO2-260 heterojunction catalysts were expected to be the promising catalyst candidates of CO oxidation for the future application. Please kindly find the revised manuscript as a reference. Thank you.
- On page 2, there is a different font style has been used. Authors should use the same font style throughout the manuscript.
Answer: Thank you very much for your constructive comments. According to your comments, the authors have corrected the errors and examined in detail the full text. Please kindly find the revised manuscript as a reference. Thank you.
- Authors show the XPS peak fitting areas in Table 3 and Table 4. However, it does not give any meaningful information. It’s better to calculate the relative percentages of different oxidation states.
Answer: Thank you very much for your constructive comments. According to your valuable suggestions, the authors calculated the relative percentages of different oxidation states. As shown in Table 3, the relative percentages of different oxidation states in Cu2O/α-MnO2, Cu2O-CuO/α-MnO2-240, Cu2O-CuO/α-MnO2-260, Cu2O-CuO/α-MnO2-280 and CuO/α-MnO2-500 catalysts were 0, 2.07, 4.11, 2.35, and ∞, respectively. These results indicated that the calcination temperature greatly affected the relative percentages of different oxidation states over the catalyst surface. Meanwhile, the XPS results also revealed the formation of the Cu2O-CuO heterojunction structure on the surface of the Cu2O-CuO-T catalyst, which was well consistent with the FTIR and XRD characterization results. In addition, the number of the Oads mainly depended on the number of oxygen vacancies at the surface. The reason is that the Oads only could be absorbed on the oxygen vacancies. According to previous reports (Chemical Reviews, 2006, 106(10): 4428-4453.), increased adsorption of ambient oxygen by surface oxygen vacancies will lead to the formation of oxygen anion radicals, which will further improve the performance of the catalyst. Thus, the Oads/(Olatt + Oads) ratio became the valid parameter to analyze the content of the surface oxygen vacancy. Please kindly find the revised manuscript as a reference. Thank you.
Table 3. Cu 2p peak area of the as-prepared Cu2O/S (S = α-MnO2, CeO2, SBA-ZSM-5, Fe2O3), Cu2O/α-MnO2, Cu2O-CuO/α-MnO2-T and CuO/α-MnO2-500 catalysts.
Samples |
Cu2O peak area |
CuO peak area |
Satellite Peak area |
|||
Cu2O/CeO2 |
58985.93 |
- |
- |
0 |
||
Cu2O/Fe2O3 |
23117.54 |
- |
- |
0 |
||
Cu2O/ZSM-5 |
15803.46 |
- |
- |
0 |
||
Cu2O/α-MnO2 |
28836.83 |
- |
- |
0 |
||
Cu2O-CuO/α-MnO2-240 |
8059.77 |
16647.28 |
33992.70 |
2.07 |
||
Cu2O-CuO/α-MnO2-260 |
8803.58 |
36178.22 |
49118.25 |
4.11 |
||
Cu2O-CuO/α-MnO2-280 |
7076.77 |
16647.28 |
54040.24 |
2.35 |
||
CuO/α-MnO2-500 |
- |
55346.21 |
96344.52 |
∞ |
||
- How were the dispersion patterns of CuOx on these supports? Atomically dispersed? Or in the form of clusters or particles? The dispersion of Cu should be measured in order to support the activity.
Answer: Thank you very much for your valuable and inspiring comments. Your comments are greatly reasonable. The authors analyzed the phase composition and dispersion of the active components of the catalyst by XRD characterization. Figure S1 displayed the XRD patterns of the as-prepared Cu2O/S (S = α-MnO2, CeO2, ZSM-5, Fe2O3) catalysts with 15% Cu2O loading amount. As shown in the Figure S1, the diffraction peaks of Cu2O/S catalysts were mainly situated at 2θ = 36.6°, 42.5°, 61.7° and 73.7°, which could be ascribed to the (111), (200), (220) and (311) of Cu2O phase (PDF-#-05-0667). However, the peaks intensities of the Cu2O peaks on these catalysts were quite different. Specifically, the peaks intensities of the Cu2O peaks on Cu2O/α-MnO2 and Cu2O/CeO2 catalysts were much weaker than that on Cu2O/ZSM-5 and Cu2O/Fe2O3 catalysts, suggesting the high dispersion of the Cu2O species. This indicated that the properties of the support could greatly affect the dispersion of Cu2O, which was further conducive to the improvement of the catalytic activity of CO oxidation. We sincerely wish that our explanation can make you satisfied and clear. Thank you.
Figure S1. X-ray diffraction patterns of the as-prepared Cu2O/S (S = ZSM-5, CeO2, α-MnO2 and commercial Fe2O3) catalysts.
- It is very surprising to see that there is no XPS Cu 2p 1/2 peak in Cu2O/α-MnO2 catalysts (Fig. 11 and Fig. 12). This is completely wrong. Authors should repeat the XPS analysis of Cu 2p in all samples.
Answer: Thank you for pointing out the mistakes in the manuscript. We are very sorry for our negligence of Cu 2p 1/2 peak in Cu2O/α-MnO2 catalysts. The authors reviewed the original data and found that the XPS spectra of all the catalysts had Cu 2p ½ peaks, but we neglected the Cu 2p ½ peaks of Cu2O/α-MnO2 catalysts. Generally, there were two main peaks around 933.08-933.82 eV and 952.68-952.88 eV observed over these catalysts Fig. 11 (a) and Fig. 12 (a), which could be attributed to the Cu 2p 3/2 and Cu 2p 1/2 peaks, respectively. In addition, the authors have already reviewed the XPS analysis of Cu 2p in all samples according to your valuable suggestions. Please kindly find the revised manuscript as a reference. Thank you.
- Authors should record the XPS Mn 3p to differentiate the Mn oxidation states. The current Mn 2p spectra do not give any meaningful information.
Answer: Thank you very much for your inspiring and sensible suggestion. Your concern is quite reasonable. In fact, the authors also plan to add XPS Mn 3p spectra, but the XPS results are not available yet due to the pandemic of COVID-19 and the close of the lab during the summer vacation. The authors sincerely apologize for this. Thank you very much for your kind understanding. The chemical state analysis of Mn cannot simply analyze its Mn2p peak, although the peak difference between elemental Mn and oxide Mn2p is more, which can be well separated. However, the difference of Mn2p peak of different valence oxides of Mn is relatively small, and other methods are needed for auxiliary identification, such as 3p peak of Mn or 3s peak of Mn. Your inspiring perspective will be greatly helpful and useful in our future research. Thank you.
- It is well known that the discrimination of different Cu oxidation states by XPS cannot be easily performed; a comparison with Auger parameters should be performed (Catal. Sci. Technol., 2021, 11, 6134-6142).
Answer: Thank you very much for your constructive and inspiring comments. The authors previously did not perform the Auger spectrum collection on the elements during the XPS analysis. The authors only analyzed the samples by XPS photoelectron spectroscopy; therefore, the LMM parameters of the elements could not be obtained. In addition, it could be concluded the states of the Cu species (Cu2O and CuO) in all catalysts based on XRD and XPS analyses of the catalysts. Thank you.
- On page 14, very small labels in Figure 12 are difficult to read, and the figure quality needs to be improved.
Answer: Thank you very much for pointing out the weakness in manuscript. Following your valuable suggestions, the authors magnified the labels in Figure 12, which made the labels easily readable. Besides, the authors also devote the best efforts to revise the whole manuscript to avoid the possible mistakes. Please kindly check the revised manuscript as the reference. Thank you.
- The title speaks about “The Roles of the Support Type and Cu2O-CuO Interface Effect”. There is no evidence of the involvement of the Cu2O-CuO interface effect.
Answer: Thank you very much for your attention to our manuscript. In this work, a series of Cu2O/S (S = α-MnO2, CeO2, ZSM-5, and Fe2O3) supported catalysts with Cu2O loading amount of 15% were prepared by the facile liquid phase reduction deposition-precipitation strategy and investigated as the CO oxidation catalysts. It was found that the Cu2O/α-MnO2 catalyst exhibited the best catalytic activity for CO oxidation. Besides, a series of Cu2O-CuO/α-MnO2 heterojunction with varied proportion of Cu+/Cu2+ were synthesized by further calcining the pristine Cu2O/α-MnO2 catalyst. As could be observed in Figure. 2, the main trend of CO conversion on these catalysts was to gradually increase as the reaction temperature increased until 100% CO conversion was reached. In addition, it was worth noting that the formation of Cu2O-CuO heterojunctions greatly improved the catalytic activity of the catalyst. The only difference between Cu2O/α-MnO2, Cu2O-CuO/α-MnO2-T and CuO/α-MnO2-500 catalysts was the change of heterostructure. This also reflected the involvement of Cu2O-CuO interface effect from the side. The authors sincerely wish that the explanation can make you satisfied and clear. Thank you.
Figure 2. The curves of the CO conversion versus reaction temperature over the as-prepared Cu2O/α-MnO2, Cu2O-CuO/α-MnO2-T and CuO/α-MnO2-500 catalysts; reaction conditions: CO/O2/N2 = 1/20/79, GHSV = 12000 mL g-1 h-1, 1 atm.

Round 2
Reviewer 2 Report
The current version of the manuscript need further revisions. The author's may need address the following questions.
1. In the current spectra, removed the experimentally observed Cu 2p peak. Fig. 11a and Fig. 11b XPS spectra looks very unusual. The authors must include the original Cu 2p peak while doing the peak fitting. 2. The author's show the XPS peak fitting areas in Table 3 . However, it does not give any meaningful information. It’s better to calculate the relative percentages of different Cu oxidation states and include in the table 3.Author Response
The current version of the manuscript needs further revisions. The authors may need address the following questions.
Answer: Thank you very much for providing such constructive and valuable suggestions to us. The authors are very sorry that previous replies to the review opinions did not meet your requirement. The author all agreed that your valuable opinions were of great significance to improve the manuscript quality. Herein, the authors revised the article according to your advisable comments. Please kindly find the revised manuscript as a reference. The authors sincerely wish that the revised manuscript could make you satisfied. Any further suggestions as well as comments are also greatly welcome and appreciated.
- In the current spectra, removed the experimentally observed Cu 2p peak. Fig. 11a and Fig. 11b XPS spectra looks very unusual. The authors must include the original Cu 2p peak while doing the peak fitting.
Answer: Thank you very much for your constructive comment and suggestion. Following your valuable advice, the authors have added the original Cu 2p peak while doing the peak fitting. Please kindly find the revised manuscript as the reference. Thank you.
- The author's show the XPS peak fitting areas in Table 3. However, it does not give any meaningful information. It’s better to calculate the relative percentages of different Cu oxidation states and include in the table 3.
Answer: Thank you very much for your constructive comment and inspiring suggestion. Following your valuable advice, the authors have calculated Cu2O peak area ratio, CuO peak area ratio and the relative percentages of Cu2+/Cu1+. The results are summarized in Table 3. Please kindly find the revised manuscript as the reference. Thank you.

Round 3
Reviewer 2 Report
The authors’ responses and revisions have satisfactorily addressed my comments on the earlier version of the manuscript.